# Study team perspectives on a multisite randomized clinical trial with underserved rural populations: A mixed methods feasibility analysis

Kristina L. Foster[1]*◉, Sarah Sanders[2]◉, Christi A. Madden[3], Lacy Malloch[4], Amy Swango-Wilson[5], Barbara Jandasek[6], Maryam Y. Garza[7], Jaime Baldner[8], Erin Dawley[9], Timothy VanWagoner[10], Jerome Philip Saul[11], Ann M. Davis[1]

1 Department of Pediatrics, University of Kansas Medical Center, Kansas City, Kansas, United States of America, 2 Department of Pediatrics, University of New Mexico, Albuquerque, New Mexico, United States of America, 3 Translational Research Institute, University of Arkansas for Medical Sciences, Little Rock, Arkansas, United States of America, 4 Department of Pediatrics, University of Mississippi Medical Center, Jackson, Mississippi, United States of America, 5 Department of Research Services, Alaska Native Tribal Health Consortium, Anchorage, Alaska, United States of America, 6 Department of Psychiatry and Human Behavior and the Department of Pediatrics, Warren Alpert Medical School of Brown University, Bradley Hasbro Children's Research Center, Brown University Health, Both in Providence, Rhode Island, United States of America, 7 Department of Population Health Science Center, University of Texas Health Sciences at San Antonio, San Antonio, Texas, United States of America, 8 Department of Biomedical Informatics, University of Arkansas for Medical Sciences, Little Rock, Arkansas, United States of America, 9 Department of Pediatrics, Medical University of South Carolina, Charleston, South Carolina, United States of America, 10 Department of Pediatrics, University of Oklahoma Health College of Medicine, Oklahoma City, Oklahoma, United States of America, 11 Department of Pediatrics, West Virginia School of Medicine, Morgantown, West Virginia, United States of America

◉ These authors contributed equally to this work.
* kfoster6@kumc.edu

## Abstract

Although the literature on clinical trial methodology is quite robust, the voices of study staff as key influencers of this process are lacking, particularly for rural and underserved pediatric clinical trials. Using qualitative and quantitative (i.e., survey) methodology, the purpose of the current study was to gather information from study investigators and staff who served on one of the initial multi-state trials in the IDeA States Pediatric Clinical Trials Network (ISPCTN) regarding barriers and facilitators of conducting this rural clinical trial. Quantitative analysis indicated most study investigators and staff who responded (55%) were neutral about the various recruitment methods. Qualitative analyses identified 6 relevant themes: 1) Participant families felt overwhelmed with study procedures, 2) Incentives are important and should be given in a timely fashion to child as well as adult participants, 3) A personal connection is key to engagement and retention, 4) Specific recruitment materials and methods are preferred including family friendly consent forms and advertisements that clearly explain study procedures and clear expectations, 5) There was enthusiasm for the intervention and ideas for consideration in implementing future interventions of this type, and 6) Staff expressed enthusiasm for working in rural areas with rural

**Data availability statement:** The data for the qualitative part of the study described in this manuscript are contained within the manuscript. The data from the iAmHealthy study are publicly available in NICHD DASH at https://dash.nichd.nih.gov/.

**Funding:** This work was funded by grants from the National Institutes of Health: UG1 OD024943 (KF, AD) UG1 OD024947 (SS) UG1 OD024950 (CM, TV) UG1 OD024942 (LM) UG1 OD024944 (ASW) UG1 OD024951 (BJ) U24 OD024957 (MG, JB) UG1 OD024956 (ED) UG1 OD030016 (JS) The funders had no role in study design, data collection and analysis, decision to publish, or preparation of the manuscript.

**Competing interests:** The authors have declared that no competing interests exist.

participants and appreciated the unique aspects of working with this population. This paper provides valuable insight into the operational feasibility of a large, multi-site behavioral intervention trial and outlines lessons learned from study personnel with actionable tips for improving recruitment, retention, and other study procedures. These staff are open to various recruitment methods, and are enthusiastic about working with underserved, rural families. They report that they believe families can be overwhelmed by study procedures, and that a personal connection with families can facilitate study conduct.

## Introduction

Conducting randomized controlled clinical trials in rural and underserved pediatric populations in geographically diverse areas presents unique challenges. The National Institutes of Health formed a pediatric research network, called the Environmental Child Health Outcomes IDeA States Pediatric Clinical Trials Network (ECHO ISPCTN) that is designed to facilitate the delivery of state-of-the-art clinical trials to rural or underserved children and families [1]. Despite the support of this network, barriers to recruitment, retention, and other study activities remain.

Recruitment is the top barrier to the successful completion of clinical trials [2], and studies involving rural and underserved families face even greater recruiting challenges [3]. Experts categorize barriers to recruitment as either operational/administrative or participant-level. Operational and administrative barriers include the lack of translation services [4]; research training not connecting knowledge to practice [4,5]; lack of early engagement or ongoing relationships with the community or clinical staff by the research team [6–10]; delay between study training and recruitment resulting in lost momentum [8]; clinician time constraints [6,11,12]; fewer than expected or competition for available eligible participants [11,13]; strain on clinical resources and space [6,11]; and the need to be flexible and modify recruitment strategies [10,13]. Participant-level barriers include demographic factors such as rural, racial and ethnic minorities being skeptical of trial participation [8,14], as well as differing perceived seriousness of the relevant medical condition [8,13,14].

Although many experts agree that it is important to include rural and underserved families in clinical trials [1], including these special populations can be challenging beyond recruitment [15]. Studies struggle to retain rural families, and previous research has focused on improving retention rates for rural participants with specific approaches [16] such as repeated contacts [17]. Crocker and colleagues found that long periods between follow-up visits can be problematic [12], and Long et al. reported that participants lost interest in attending follow-up visits once the visits were thought to be unnecessary. They suggested mitigating this phenomenon by linking research activities to routine clinical visits [8].

The existing literature to date has primarily focused on participant perspectives as they are key to a successful clinical trial. However, input from clinical trials coordinators and other research staff is also important as they have valuable insights on

barriers and facilitators to the clinical trials process. Previous research with clinical trials coordinators and other research staff has been limited to a focus on recruitment [13,18], but suggests that staff prefer to be involved early in the research process (i.e., during protocol development), and that research staff are often anxious about various study tasks which can interfere with study completion [13]. Specific to trials with rural and underserved children, a recent scoping review [19] found that most of the existing research on rural pediatric clinical trials consisted primarily of descriptions of study strategies, with limited research on the perspectives of study staff regarding the conduct of clinical trials with rural and underserved children.

The current paper aims to address this gap in the existing literature by using a mixed methods approach to assess study investigator and staff perspectives regarding the conduct of a multi-state randomized controlled clinical trial with rural children and their caregivers. Specifically, we report on the viewpoints of the staff for Feasibility Trial of the iAmHealthy Intervention (NCT04142034), one of the first behavioral intervention clinical trials conducted in the ECHO ISPCTN.

## Methods

The iAmHealthy Feasibility Study [20], hereafter referred to as iAH was overseen by the ISPCTN Data Coordinating and Operations Center (DCOC) at University of Arkansas for Medical Sciences and their IRB acted as the central IRB for all sites. The clinical trial engaged 6 ISPCTN sites: Institutions in Delaware, Nebraska, South Carolina, and West Virginia served as clinical study sites and an institution in Oklahoma served as the study coordination site with the Kansas site leading the intervention team. The iAH trial was a pilot study that randomized recruitment methods at the site level (Consecutive, Traditional), and randomized participants at the individual level (iAH vs Control).

The study recruited children seen in participating rural clinics who were 6–11 years of age with a BMI > 85$^{th}$ percentile and their primary caregiver. Caregivers signed written informed consent, and the children signed written assent. Clinic rurality was determined with the RUCA ZIP Code crosswalk version 3.1 available from the University of North Dakota [21]. The primary objective of the iAH trial was to assess key variables such as: participant recruitment, participant retention, intervention dose, and blinding. The iAmHealthy intervention was delivered via interactive televideo to caregivers and children, and included topics such as child exercise without peers, eating at social/group gatherings, increased attention to self-esteem and decreased focus on eating fast-food. The intervention is composed of 26 contact hours: 15 hours of group sessions and 11 hours of individual sessions. Groups met weekly for 12 weeks, followed by monthly for 3 months, for a total intervention period of 6 months. Baseline and post measures conducted remotely included height and weight, 3-day diet-recall and 7-day physical activity monitoring by accelerometer. The total possible compensation for iAH participation was $240: $20 consent, $20 for each month of the 6-month intervention period ($120), and $100 for completing post-intervention assessments and returning equipment. Enrolling coordinators were blinded to which group the participant was assigned. From there, the unblinded coordinator team stayed connected throughout the intervention period. The enrolling (blinded) coordinator did not have contact with participant again until the end of the study when they ensured post-measures were conducted and equipment returned. Primary outcome data from the trial were published elsewhere [22] and indicated that the vast majority of patients were recruited using the active method of recruitment (95%); data not yet published indicate that 91% of the sample was retained through the final measurement.

At the conclusion of the trial, study staff at each of the 6 sites and those involved in the intervention delivery or other aspects of the protocol were invited to participate in the current study which included 2 questionnaires. Study staff included site principal investigators, study coordinators, and study support staff from each site (referred to henceforth as "study-site personnel"), as well as staff involved in the delivery of the behavioral intervention or the collection of measures (referred to henceforth as "non-study site personnel"). Both questionnaires were distributed using the REDCap [23,24] web application managed by the DCOC shortly after trial activities concluded. The surveys were designed by study

leadership to elicit information from all study personnel on their overall satisfaction with recruitment options, intervention arms, and the overall feasibility of the iAH study.

It is important to note that the iAH study began recruitment February 3, 2020, with some of the first activities taking place in person then rapidly transitioning to remote procedures with the outbreak of the Coronavirus Disease 2019 (COVID-19) pandemic. End of recruitment was June 7th, 2020, and study completion was March 25, 2021.

### Survey 1: Recruitment

In addition to site and study role, Survey 1 involved 8 questions (Table 1); one that asked for a numerical rating of the three recruitment methods (Consecutive/Retrospective, Consecutive/Prospective, Traditional), and others that solicited open-ended comments regarding recruitment activities. The Consecutive/Retrospective method involved recruiting patients with healthcare visits from the prior year prescreened for likely eligibility and contacted first via letter, then phone. The Consecutive/Prospective method involved recruiting participants who were prescreened for likely eligibility during their healthcare visits. The Traditional method involved recruiting participants who self-identified after seeing flyers, emails, social media, and newsletters.

### Survey 2: Overall study conduct

In addition to site and study role, Survey 2 involved 8 qualitative questions (Table 2) on retention and study procedures.

## Results

Descriptive statistics were used to summarize quantitative results. For qualitative analyses, free text in both surveys was analyzed using the Morgan and Krueger [25] approach which focused on thematic analysis. Each coder (KF, BJ, CM, SGS, ASW) reviewed responses individually to generate an initial code set and then coded the data. Afterwards, coders participated in a series of meetings with an experienced qualitative researcher (AD) to discuss individual coding decisions and iteratively determined themes using a consensus-driven approach. As themes were defined and finalized, earlier

**Table 1. Survey 1.**

| Survey 1: Numeric Rating of Recruitment Methods | |
|---|---|
| 1. On a scale of 1–5, where 1 = Not at all difficult and 5 = Extremely difficult, please rate each recruitment method based on its level of difficulty? | |
| | a. Recruitment Option 1- Consecutive: Retrospective |
| | b. Recruitment Option 1- Consecutive: Prospective |
| | c. Recruitment Option 2- Traditional |
| **Survey 1: Qualitative** | |
| 2. What are your thoughts on the recruitment methods used in the study? | |
| 3. Do you have any suggestions to make recruitment easier? | |
| 4. Do you feel you needed more support throughout the recruitment process from the DCOC or study team leadership? | |
| 5. Do you have any ideas on what would have made recruitment more acceptable to the participants? | |
| 6. Is there anything you would have done differently? | |
| 7. Is there anything you wish we (DCOC/study leadership) would have done differently? | |
| 8. Do you have any additional feedback or comments? | |

**Table 2. Survey 2.**

| Survey 2: Qualitative |
| --- |
| 1. Do you have any suggestions to make retention easier? |
| 2. Do you feel you needed more support throughout the intervention period (post recruitment) from the DCOC or study team leadership? |
| 3. Do you have any ideas on what would have made the study more acceptable to the participants? |
| 4. What did you like best about the study? |
| 5. Is there anything you would have done differently? |
| 6. Is there anything you wish we would have done differently? |
| 7. Did you get a copy of the iAmHealthy curriculum or newsletters? |
| a. If yes, have you used these materials outside of the iAmHealthy program? |
| 8. Do you have any additional feedback or comments? |

coded responses were reassessed and recoded or re-categorized as needed to ensure consistency. All coders agreed that thematic saturation was reached and then quotes to support the final set of themes were identified. These preliminary results were presented to the full research team for review and opportunity for feedback. Consensus was reached amongst the group in determining the final themes and quotes.

Survey 1 had a 90% response rate (27 of 30 participants), and Survey 2 had an 83% response rate (25 of 30 participants). Table 3 describes the breakdown by role of the personnel that participated in each survey. Study support and regulatory staff had the lowest response rates (2 of 4 and 1 of 4 for Survey 1 and Survey 2 respectively). Quantitative results (Table 4) indicate most respondents were neutral about all recruitment methods (Mdn = 3, IQR = 1.0–1.5). Fourteen

**Table 3. Roles of survey participants.**

| | Study-Site Personnel[1] | | | Non-Study Site Personnel[2] | | | Total (n = 30) |
| --- | --- | --- | --- | --- | --- | --- | --- |
| | Investigators (n = 4) | Coordinators (n = 9) | Other[3] (n = 4) | Unblinded Coordinators (n = 4) | Intervention Team (n = 8) | Other[4] (n = 1) | |
| Survey 1 | 4 | 8 | 2 | 4 | 8 | 1 | 27 |
| Survey 2 | 3 | 9 | 0 | 4 | 8 | 1 | 25 |

[1]Study site personnel located at one of the 5 ISPCTN sites.

[2]Non-study site personnel were not directly affiliated with any of the 5 ISPCTN sites but carried out study-related activities including intervention delivery and conduct of assessments.

[3]Study support staff and regulatory staff.

[4]Member of the dietary recall team.

**Table 4. Quantitative findings.**

| | Recruitment Option 1- Consecutive: Retrospective Median (IQR) | Recruitment Option 1- Consecutive: Prospective Median (IQR) | Recruitment Option 2- Traditional Median (IQR) |
| --- | --- | --- | --- |
| All respondents (n = 27) | 3 (1.5) | 3 (1.0) | 3 (1.0) |
| Principal Investigators (n = 5) | 1 (2.0) | 2 (0.0) | 2 (1.0) |
| *Coordinators (n = 8) | 2 (2.0) | 2 (2.0) | 2.5 (2.5) |

Question asked: On a scale of 1–5, where 1 = Not at all difficult and 5 = Extremely difficult, please rate each recruitment method based on its level of difficulty.

*Coordinators who were directly involved with study recruitment.

respondents reported that they did not participate in recruitment but still rated the recruitment options, raising questions about the validity of responses to this question. Primary investigators (n = 5) found the Consecutive Retrospective option to be not at all difficult (Mdn = 1, IQR = 2.0) and the other two options to be slightly difficult (Mdn = 2, IQR = 0.0–1.0). Coordinators that recruited participants considered Consecutive Retrospective and Consecutive Prospective slightly difficult (Mdn = 2, IQR = 2.0) while the median for the Traditional method was slightly higher (Mdn = 2.5, IQR = 2.5).

Regarding qualitative analyses, six major themes were identified and reached saturation, with specific quotes supporting each theme presented in Table 5.

**Table 5. Qualitative themes and supporting quotes.**

**1. Families felt overwhelmed by study procedures.**

**"I feel like a lot of the families got overwhelmed with the amount of tasks that were needed to be complete in a short period of time."**

**"It probably caused one of our patients to drop out in the end because she was overwhelmed with the calls. I would suggest not trying to cram all of those activities in one month at the end. Maybe allow a longer window for the final. Parents are NOT wanting to take kids out of school, so we were lucky to get those in person measurements in the time frame available."**

**"Less phone calls to the families overall. It was confusing for families who was who when it came it the blinded coordinators, the unblinded coordinators, and the diet recall team members calling them all different days of the week."**

2. Incentives are important, should be given in a timely fashion to child as well as adult participants.

"We did not provide timely compensation to participants on the schedule that was promised. In some cases, this caused families to consider withdrawing."

"Also provide an incentive to the CHILD along the way such as a water bottle, a pedometer, or some small token with their newsletters with the I Am Healthy Logo each month just to keep them engaged. Kids love to 'get things'. Many of them talked about something they learned about 'red foods' so something or some monthly gift from the team that reinforces the lesson or newsletter would have been great."

"Possibly offer more incentives for the families throughout the study. Small incentives that they can look forward too."

3. A personal connection is key to engagement and retention.

"The participants were recruited by each site, and they had a particular connection with those who did the visits, recruited them, paid them, etc. They had a personal connection to them. Once they were turned over to someone else in another state, the connection was lost…We recruited them, cultivated them, and so I think we might have had better retention if we 'kept' them."

"I would have kept closer track of the way that families were falling behind on health coaching hours. By the time we identified this problem, it required some pretty intense effort to correct the issue."

"I think all forms of recruitment were beneficial. It may have helped to make recruitment more integrated by establishing the PCP patient encounter to include a warm hand off (while the patient is in the exam room) to the site recruiter. The more integrated the service, the more likely the family has a sense of trust."

4. Specific recruitment materials and methods are preferred including family friendly consent forms and advertisements that clearly explain study procedures and clear expectations.

"Easier to understand and more family-friendly consent forms and advertisements."

"More printed information and local advertisements/mailers to families/more printed resources to send families before calls would have been very helpful."

"The variety of recruitment methods, along with being able to text families is what really allowed us to be successful and reach more potential participants overall."

5. There was enthusiasm for the intervention and ideas for consideration in implementing future interventions of this type.

"I think weekly group calls is ideal for retention and routine. I saw a drop off in some participants when we switched to once a month group calls."

"I very much appreciated the opportunity to participate in this project. I thought it was well conducted and directed. From the perspective of intervention delivery, the families seemed engaged with the curriculum."

"In my opinion, the group exercise activities that started in late December were a big benefit to keep families on a schedule. I think families are looking for success in a program like this, and structure helps a lot."

6. Staff expressed enthusiasm for working in rural areas with rural participants and appreciated the unique aspects of working with this population.

"I liked that it was reaching families who really do have limited resources. These families were really thankful for a study like this in their rural area, as opposed to having to drive over an hour to see weight management."

"It was a good experience, especially for [REDACTED], to bring research to the rural areas and families. I like how we got to reach and help families who were having a hard time finding these type of resources in their area. Oddly enough, doing things remotely was very convenient for families."

"…It was rewarding to help someone make lifestyle changes for themselves and their family in a virtual setting where many of the participants would have not been able to attend a clinic and during COVID were able to do this in a safe manner."

**Theme #1**

Families felt overwhelmed by study procedures. Respondents indicated that study procedures like phone calls and study tasks seemed overwhelming and burdensome to participants. Phone calls to participants came from several different members of the study team multiple times over the course of the trial. These included blinded coordinators managing initial recruitment and enrollment communications, unblinded coordinators and the diet recall staff each calling participants on a weekly or monthly schedule, and contacts from intervention staff. The one-month end of study data collection window was perceived as too narrow. Because of the outbreak of the COVID-19 pandemic, many planned in-person recruitment and enrollment procedures were moved online [26]. Staff reported that these online remote protocols added a layer of additional communication demands on study staff and participants to collect baseline and end-of-study data.

**Theme #2**

Incentives are important and should be given in a timely fashion to child as well as adult participants. Survey respondents noted the importance of adequate and timely participant compensation. They indicated that iAH participants expressed that they were not paid enough for study procedures. Survey respondents indicated that they thought providing incentives to the children during the intervention would have increased engagement and had the added benefit of reinforcing the lessons. Several respondents mentioned that lack of timely participant compensation payments was problematic. Suggestions were provided to streamline payment processes, such as paying participants once at the beginning and once at the end.

**Theme #3**

A personal connection is key to engagement and retention. The importance of personal connection between study staff and research participants for study retention and intervention adherence was noted as important and yet some study procedures prevented the recruiting coordinators from maintaining these relationships with the participants. Once a family was recruited and enrolled, their interactions with study staff were limited to the unblinded coordinators/non-study-site personnel until the post intervention assessment period. The recruiting coordinators were advised to limit contact to protect the study blind. Respondents also expressed that a connection with referring healthcare providers during the recruitment period is key. One respondent suggested that having a provider introduce their patients to the study recruiter may have been beneficial. Finally, if it had been allowed by the protocol, the personal connection created during the enrollment process between the research participants and the enrolling study staff could have promoted adherence to the study intervention through improved attendance at the individual and group sessions.

**Theme #4**

Specific recruitment materials and methods are preferred, including family friendly consent forms and advertisements that clearly explain study procedures and expectations. Respondents reflected that it is necessary to provide clear explanations of study procedures and expectations to successfully enroll participants. Consent forms and advertisements written at the participant level of understanding were identified as important avenues to successful recruitment. Staff indicated they would have liked to have better ways to explain the study during recruitment. One suggested using a video to describe the study and stimulate interest. The use of a variety of recruitment methods and communication styles were viewed as important, including fostering a "general awareness through media."

**Theme #5**

There was enthusiasm for the intervention with ideas for consideration in future interventions of this type. Respondents indicated high enthusiasm for the intervention and proposed many ideas to consider for future interventions of this type.

There was a suggestion to maintain weekly group iAH intervention calls for the entirety of the behavioral intervention period instead of switching to monthly calls, contrasting with the idea above that the overall amount of study activities seemed overwhelming for participants. Also, respondents voiced a desire for more detail around the boundaries of various intervention staff roles and suggested it would be helpful to add this level of detail to the manual of procedures.

**Theme #6**

Staff expressed enthusiasm for working in rural areas with rural participants and appreciated the unique aspects of working with this population. Many staff wrote about their enthusiasm for engaging with rural participants. They found that it was rewarding and "truly enjoyed" reaching families that often lack resources due to their geographic location. Staff also noted that the changes to study procedures made in response to the COVID-19 pandemic rendered study participation more feasible for rural participants, i.e., the change from in-person to virtual study activities. This shift in research procedures for COVID-19 risk mitigation was ultimately seen as a net positive by respondents.

## Discussion

The purpose of the current paper was to use a mixed methods approach to assess study staff perspectives regarding the conduct of a multi-site pediatric randomized controlled clinical trial with rural or underserved children and their caregivers. The iAH study was a first of its kind clinical trial evaluating the feasibility and effectiveness of different recruitment strategies. The study found that sites reached full enrollment using the active (consecutive) method and no sites had success with traditional recruitment [22]. However, our quantitative survey results indicated study staff had no preferences among Consecutive (Prospective, Retrospective) and Traditional recruitment approaches, with all recruitment approaches receiving similar ratings. This is surprising as the three methods were not equivalent in terms of effort. For the traditional method, simple activities such as posting flyers or sharing an announcement on Facebook were implemented, which required much less effort than calling families who had been recently seen in the clinic (retrospective) or approaching the families when they were in the clinic for a visit (prospective). Other studies indicate, for example, that calling families or approaching them in clinic can be anxiety provoking for coordinators [13].

Effective recruitment strategies vary by study, population and other factors. Additionally, with the move since COVID-19 to more remote study visits and activities, the challenge of study enrollment has only increased in complexity. Finding successful and practical strategies that are not onerous on study staff are powerful tools in a site's recruitment toolkit. Study staff without direct recruitment responsibilities seem to have enough knowledge of recruitment processes to form an opinion about how those efforts are going. They are involved in study meetings and likely hear how recruitment is going and work closely enough with those who are recruiting to draw their own conclusions. While their opinions may not be the most relevant, they still have value. Additionally, while these study staff answered the quantitative question about recruitment, they did not answer the qualitative portion.

Qualitative findings indicate that study staff believed families felt overwhelmed by study procedures wanted larger and more timely payment for study incentives and felt that a personal connection between themselves and families was important to successful study retention. Regarding the perception of participant burden, previous research suggests a disconnect between participant and staff perceptions. Shilling and colleagues [27] found that the research staff concerns that parents would be burdened by being approached to participate in research were largely unfounded, with most parents viewing the chance to participate as an exciting opportunity. Existing research suggests that the perception of study-related burden may differ by race, with those who identify as white more likely to decline participation due to perceived burden, and those who identify as Black more likely to decline due to lack of interest or family pressures [28]. Awareness of ongoing trials is also a major limiting factor to participation in clinical trials among underserved communities [3], but by using the active Consecutive recruitment methods (Prospective, Retrospective), the iAH study was able to overcome this issue [22].

Previous research with medical and research professionals supports the importance of building personal connections to successful recruitment and suggests specific behaviors that can promote positive relational interactions including "listening to personal information, expressing empathy, and then providing reciprocal self-disclosures." [17]. Study staff enthusiasm for both the iAH behavioral intervention, and for working with underserved, rural populations are factors that may have facilitated personal connections with participants.

The current study is not without limitations. It was limited to 6 sites, which may not be representative of all sites. Second, due to staffing changes at 1 of the 6 sites, responses to questionnaires do not equally represent each site. Third, the study procedures described herein (various types of recruitment, remote behavioral intervention) may be limited to research groups conducting similar studies. Finally, because the staff who participated in this project were part of a national pediatric clinical trials network (the ECHO ISPCTN) their opinions may be significantly different than staff who are not part of a supported network.

## Conclusions

Study investigators, coordinators and other research staff are critical to the conduct of clinical trials with rural and underserved children and families and future research is needed to raise the voices of these key partners in clinical trials conduct. These staff are often open to various recruitment methods, and are enthusiastic about working with underserved, rural families. They report that they believe families can be overwhelmed by study procedures, and that a personal connection with families can facilitate study conduct, both of which should be addressed in future studies. These findings could have important implications for investigators hoping to bring pediatric clinical trials to rural and underserved families.

## Acknowledgments

We would like to acknowledge and thank the many wonderful research staff in our network who participated in this project.

## Author contributions

**Conceptualization:** Kristina L. Foster, Sarah Sanders, Timothy VanWagoner, Ann M. Davis.

**Data curation:** Kristina L. Foster, Sarah Sanders, Christi A. Madden, Jerome Philip Saul, Ann M. Davis.

**Formal analysis:** Sarah Sanders, Amy Swango-Wilson, Barbara Jandasek, Erin Dawley, Timothy VanWagoner, Jerome Philip Saul, Ann M Davis.

**Investigation:** Maryam Y. Garza.

**Methodology:** Maryam Y. Garza.

**Project administration:** Kristina L. Foster, Maryam Y. Garza, Jaime Baldner.

**Resources:** Jaime Baldner.

**Software:** Maryam Y. Garza, Jaime Baldner.

**Supervision:** Amy Swango-Wilson, Jaime Baldner, Jerome Philip Saul, Ann M Davis.

**Validation:** Maryam Y. Garza, Jaime Baldner, Ann M. Davis.

**Visualization:** Christi A. Madden, Barbara Jandasek, Maryam Y. Garza, Jaime Baldner.

**Writing – original draft:** Kristina L. Foster, Sarah Sanders, Lacy Malloch, Amy Swango-Wilson, Barbara Jandasek, Ann M. Davis.

**Writing – review & editing:** Kristina L. Foster, Sarah Sanders, Christi A. Madden, Lacy Malloch, Amy Swango-Wilson, Jaime Baldner, Ann M. Davis.

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
