## [Decision Letter · Decision Letter 0]

20 Jun 2025

PONE-D-25-24666Study team perspectives on a multisite randomized clinical trial with underserved rural populations: A mixed methods feasibility analysisPLOS ONE

Dear Dr. Foster,

Thank you for submitting your manuscript to PLOS ONE. After careful consideration, we feel that it has merit but does not fully meet PLOS ONE’s publication criteria as it currently stands. Therefore, we invite you to submit a revised version of the manuscript that addresses the points raised during the review process.

We look forward to receiving your revised manuscript.

Kind regards,

Taiwo Opeyemi Aremu

Academic Editor

PLOS ONE

Journal Requirements:

Reviewers' comments:

Reviewer's Responses to Questions

**Comments to the Author**

1. Is the manuscript technically sound, and do the data support the conclusions?

Reviewer #1: Partly

Reviewer #2: Yes

2. Has the statistical analysis been performed appropriately and rigorously? 

Reviewer #1: Yes

Reviewer #2: Yes

3. Have the authors made all data underlying the findings in their manuscript fully available?

Reviewer #1: Yes

Reviewer #2: Yes

4. Is the manuscript presented in an intelligible fashion and written in standard English?

Reviewer #1: Yes

Reviewer #2: Yes

5. Review Comments to the Author

Reviewer #1: This article is a mixed-methods analysis of the barriers/facilitators and perspectives of study-staff performing a pediatric clinical trial in rural areas. The goal of the study was to assess the efficacy of recruitment, retention, and other study procedures unique to the rural pediatric setting.

The themes that emerged from the qualitative analysis provide rich data with actionable conclusions to performing pediatric clinical trials in rural areas. One weakness is the single quantitative question in Survey 1 pertaining to the perceived "difficulty" of recruitment style. This could have been expanded to provide a nuanced response to the feasibilities of the recruitment methods.

The major finding that the study staff felt "neutral" about the recruitment methods is slightly misleading due to the one question asked with the issues stated above; in addition, it appears 14 of the 27 respondents were not actually involved in the recruitment, even though they responded to the question, leading me to question the validity of the response. Lines 287-295 could be an opportunity to address this.

Numerical data on retention would compliment the themes that centered on retention, such as what were recruitment and then retention rates for each site? Perhaps this will be published elsewhere but I think it would bolster the themes related to retention.

Overall this is a very well written paper and an important topic! Thank you for doing this work.

Reviewer #2: Foster et al have studied research staff opinions/visons on how to conduct multisite randomized clinical trials with rural populations. It is a work on a quite specific subject. It is however possible to apply the results to other trial settings as well, so it has potential to be interesting for wider research communities as well.

Overall, the present paper is well-written, and I found no unclarities or errors to be revised. The study itself and the results were reported promptly and without remarkable corrections needed. However, in the discussion and conclusions section, I found only the results repeated, and the actual discussion remained somewhat short. It should be more detailed within the article limits: study conclusions to a wider audience, their meaning within research context, and suggestions for the future research, if available. Authors’ visions of the combined qualitative/quantitative research?

6. PLOS authors have the option to publish the peer review history of their article (what does this mean? ). If published, this will include your full peer review and any attached files.

**Do you want your identity to be public for this peer review?** For information about this choice, including consent withdrawal, please see our Privacy Policy .

Reviewer #1: No

Reviewer #2: No

---

## [Author Response · Author response to Decision Letter 1]

20 Aug 2025

Thank you for the review of our manuscript. We appreciate the comments of the reviewers and are pleased to resubmit to PLOS One our manuscript entitled “Study team perspectives on a multisite randomized clinical trial with underserved rural populations: A mixed methods feasibility analysis.” We have pasted each critique below along with our updates to the manuscript for each point.

First addressing editor requirements:

1. We utilized PLOS ONE’s style templates and believe we are complying with all guidelines.

2. I copied the language about our central IRB and consent processes from the Methods section of the manuscript and put it in the Ethics Statement section, so they match.

3. All the grant information in the ‘Funding Information’ section is correct. With apologies for the original ‘Financial Disclosure’ mismatch.

4. Reference list was reviewed; there are no retractions and just one addition.

Reviewer Feedback

Reviewer #1: This article is a mixed-methods analysis of the barriers/facilitators and perspectives of study-staff performing a pediatric clinical trial in rural areas. The goal of the study was to assess the efficacy of recruitment, retention, and other study procedures unique to the rural pediatric setting. The themes that emerged from the qualitative analysis provide rich data with actionable conclusions to performing pediatric clinical trials in rural areas. One weakness is the single quantitative question in Survey 1 pertaining to the perceived "difficulty" of recruitment style. This could have been expanded to provide a nuanced response to the feasibilities of the recruitment methods.

We agree with the reviewer that it could have been possible to have more than one question about recruitment. However, as the study has already been completed this is not possible at this time.

The major finding that the study staff felt "neutral" about the recruitment methods is slightly misleading due to the one question asked with the issues stated above; in addition, it appears 14 of the 27 respondents were not actually involved in the recruitment, even though they responded to the question, leading me to question the validity of the response. Lines 287-295 could be an opportunity to address this.

This is a fair concern. To address this, we added the following to line 209: “raising questions about the validity of responses to this question”. We additionally added further discussion in lines 303-309 with “Study staff without direct recruitment responsibilities seem to have enough knowledge of recruitment processes to form an opinion about how those efforts are going. They are involved in study meetings and likely hear how recruitment is going and work closely enough with those who are recruiting to draw their own conclusions. While their opinions may not be the most relevant, they still have value. Additionally, while these study staff answered the quantitative question about recruitment, they did not answer the qualitative portion.”

Numerical data on retention would compliment the themes that centered on retention, such as what were recruitment and then retention rates for each site? Perhaps this will be published elsewhere but I think it would bolster the themes related to retention.

This is a great point. We have added the following to the manuscript on lines 157-160 in the Methods section where the prior iAmHealthy trial is described: “Primary outcome data from the trial were published elsewhere [22] and indicate that the vast majority of patients were recruited using the active method of recruitment (95%); data not yet published indicate that 91% of the sample was retained through the final measurement.”

Overall this is a very well written paper and an important topic! Thank you for doing this work.

Thank you so much! We appreciate you taking the time to review and provide valuable feedback.

Reviewer #2: Foster et al have studied research staff opinions/visons on how to conduct multisite randomized clinical trials with rural populations. It is a work on a quite specific subject. It is however possible to apply the results to other trial settings as well, so it has potential to be interesting for wider research communities as well. Overall, the present paper is well-written, and I found no unclarities or errors to be revised. The study itself and the results were reported promptly and without remarkable corrections needed.

Thank you so much also! We know your time is valuable and appreciate your review.

However, in the discussion and conclusions section, I found only the results repeated, and the actual discussion remained somewhat short. It should be more detailed within the article limits: study conclusions to a wider audience, their meaning within research context, and suggestions for the future research, if available. Authors’ visions of the combined qualitative/quantitative research?

To address this valid concern, we added to and reworked the Discussion section, some of which was included above. Rather than include it all here, we hope it is appropriate to direct you to that section, lines 285-335.

---

## [Decision Letter · Decision Letter 1]

11 Sep 2025

Study team perspectives on a multisite randomized clinical trial with underserved rural populations: A mixed methods feasibility analysis

PONE-D-25-24666R1

Dear Dr. Foster,

We’re pleased to inform you that your manuscript has been judged scientifically suitable for publication and will be formally accepted for publication once it meets all outstanding technical requirements.

Kind regards,

Taiwo Opeyemi Aremu, MD, MPH, PhD

Academic Editor

PLOS ONE

Additional Editor Comments (optional):

Reviewer #1:

Reviewer #2:

Reviewers' comments:

Reviewer's Responses to Questions

**Comments to the Author**

1. If the authors have adequately addressed your comments raised in a previous round of review and you feel that this manuscript is now acceptable for publication, you may indicate that here to bypass the “Comments to the Author” section, enter your conflict of interest statement in the “Confidential to Editor” section, and submit your "Accept" recommendation.

Reviewer #1: All comments have been addressed

Reviewer #2: All comments have been addressed

2. Is the manuscript technically sound, and do the data support the conclusions?

Reviewer #1: Yes

Reviewer #2: Yes

3. Has the statistical analysis been performed appropriately and rigorously? 

Reviewer #1: Yes

Reviewer #2: Yes

4. Have the authors made all data underlying the findings in their manuscript fully available?

Reviewer #1: Yes

Reviewer #2: Yes

5. Is the manuscript presented in an intelligible fashion and written in standard English?

Reviewer #1: Yes

Reviewer #2: Yes

6. Review Comments to the Author

Reviewer #1: Thank you for addressing my concerns! One minor wording feedback I have is on line 159, instead of saying "data not yet published," it might be more prudent to say "preliminary data" or actually include the data of 91% retention in the results section. It is confusing to say data not yet published but then "publish" it here in this sentence, if that makes sense.

Reviewer #2: Within the submission files, I did not find the article version with tracked changes. However, reading the discussion section of the article, I found it substancially improved.

7. PLOS authors have the option to publish the peer review history of their article (what does this mean? ). If published, this will include your full peer review and any attached files.

**Do you want your identity to be public for this peer review?** For information about this choice, including consent withdrawal, please see our Privacy Policy .

Reviewer #1: No

Reviewer #2: No

---

## [Editor Report · Acceptance letter]

PONE-D-25-24666R1

PLOS ONE

Dear Dr. Foster,

I'm pleased to inform you that your manuscript has been deemed suitable for publication in PLOS ONE. Congratulations! Your manuscript is now being handed over to our production team.

Kind regards,

on behalf of

Dr. Taiwo Opeyemi Aremu

Academic Editor

PLOS ONE